

# Comparative mitogenomic analyses and gene rearrangements reject the alleged polyphyly of a bivalve genus

Regina L. Cunha[1], Katy R. Nicastro[1,2,3], Gerardo I. Zardi[3], Celine Madeira[1], Christopher D. McQuaid[3], Cymon J. Cox[1] and Rita Castilho[1]

[1] Centre of Marine Sciences, CCMAR, University of Algarve, Campus de Gambelas, Faro, Algarve, Portugal
[2] CNRS, Univ. Littoral Côte d'Opale, UMR 8187 – LOG – Laboratoire d'Océanologie et de Géosciences, Université de Lille, Lille, France
[3] Department of Zoology and Entomology, Rhodes University, Grahamstown, South Africa

## ABSTRACT

**Background:** The order and orientation of genes encoded by animal mitogenomes are typically conserved, although there is increasing evidence of multiple rearrangements among mollusks. The mitogenome from a Brazilian brown mussel (hereafter named B1) classified as *Perna perna* Linnaeus, 1758 and assembled from Illumina short-length reads revealed an unusual gene order very different from other congeneric species. Previous mitogenomic analyses based on the Brazilian specimen and other Mytilidae suggested the polyphyly of the genus *Perna*.
**Methods:** To confirm the proposed gene rearrangements, we sequenced a second Brazilian *P. perna* specimen using the "primer-walking" method and performed the assembly using as reference *Perna canaliculus*. This time-consuming sequencing method is highly effective when assessing gene order because it relies on sequentially-determined, overlapping fragments. We also sequenced the mitogenomes of eastern and southwestern South African *P. perna* lineages to analyze the existence of putative intraspecific gene order changes as the two lineages show overlapping distributions but do not exhibit a sister relationship.
**Results:** The three *P. perna* mitogenomes sequenced in this study exhibit the same gene order as the reference. CREx, a software that heuristically determines rearrangement scenarios, identified numerous gene order changes between B1 and our *P. perna* mitogenomes, rejecting the previously proposed gene order for the species. Our results validate the monophyly of the genus *Perna* and indicate a misidentification of B1.

# INTRODUCTION

The order and orientation of genes encoded by animal mitochondrial genomes may be highly variable, particularly among invertebrate lineages (*Boore, 1999*; *Boore, 2006*). Multiple rearrangements involving gene reversals, transpositions, reverse transpositions, or tandem duplications with subsequent random loss have been extensively described, for

Corresponding author
Regina L. Cunha, rcunha@ualg.pt

example, in crustaceans (*Kilpert, Held & Podsiadlowski, 2012*) and gastropods (*Grande, Templado & Zardoya, 2008*). While rearrangements are relatively common in mollusks of the same genus (*Rawlings et al., 2010*), they have not yet been described at the intraspecific level in this group.

Gene order inference in mitogenomes has been traditionally determined by the "primer-walking" strategy, a method that uses universal primers to amplify long PCR fragments over which new specific primers are designed to produce contiguous sequences (*Boore, Macey & Medina, 2005*). This time-consuming method is highly effective when assessing gene order because it relies on long, and sequentially-determined, overlapping fragments. High-throughput sequencing revolutionized the mitogenomic analysis of non-model organisms by allowing the data to be collected in a faster manner (*Briscoe, Hopkins & Waeschenbach, 2016*). Nonetheless, the use of Illumina's short-length reads for gene order assessment still poses notable challenges for those working with organisms without a good reference genome (*Singhal, 2013*) due to the widespread existence of repeats and the difficulty of assembling non-overlapping smaller fragments.

Studies on the mitochondrial DNA of the brown mussel *Perna perna* have generated controversial results. For instance, the mitogenome of a Brazilian specimen produced with Illumina short-reads showed a unique gene order for the species (*Uliano-Silva et al., 2016a*), very different from congeneric species. Mitogenome comparisons between the same Brazilian specimen of *P. perna* and *P. viridis* showed seven translocations of protein-coding genes (see Fig. 4 in *Uliano-Silva et al., 2016b*). Although the monophyly of *Perna* has already been established by previous phylogenies based on the nuclear ITS1 and the mitochondrial COI gene (*Wood et al., 2007*), *P. perna* was found to be more closely related to *Brachidontes exustus* than to *P. viridis* in the study of *Uliano-Silva et al. (2016b)*. Another study based on nuclear and mitochondrial data indicated a sister relationship between two *P. perna* lineages that exist contiguously along the eastern and southern coasts of South Africa (SA) (*Zardi et al., 2007*). However, wider sampling including better coverage of the distribution of the species revealed that the two lineages although showing overlapping geographic distributions did not exhibit a sister relationship (*Cunha et al., 2014*).

The analysis of gene rearrangements in mitogenomes may improve phylogenetic inference at lower taxonomic levels in which incomplete lineage sorting is pervasive and can generate misleading gene trees but where traces of gene duplication, translocation or remoulding have not yet been completely erased over evolutionary time (*Rawlings et al., 2010*). To confirm the mitochondrial gene order in *P. perna* and phylogenetic relationships within the genus *Perna*, we used the "primer-walking" methodology and Sanger sequencing to generate the mitogenome genome of a second Brazilian *P. perna* specimen. As gene rearrangements are frequently observed in bivalves even at lower taxonomic levels (*Feng et al., 2021*; *Ghiselli et al., 2013*), we expanded on previous results producing the mitogenomes of the two SA *Perna* lineages to explore putative intraspecific gene order changes. Phylogenetic analyses included all available *Perna* mitogenomes and other Mytilidae to analyze the alleged polyphyly of the genus.

## MATERIALS AND METHODS

### Taxon sampling, amplification and sequencing

The selection of the *P. perna* specimens used in this study included one individual from each South African lineage (eastern and western) and one specimen from Santa Catarina, Brazil, (hereafter called B2) from the same geographic area as the specimen for which the mitogenome was previously sequenced (B1; NCBI GenBank accession number: KM655841). Rhodes University approved sampling of the eastern and western South African *P. perna* specimens (Research Reference Number: RES2019/30) and Federal University of Santa Catarina approved sampling of the Brazilian *P. perna* specimen under the MTA – Material Transfer Agreement, SISGEN (https://sisgen.gov.br/) registration number: RB435EFp. All specimens were preserved in 96% ethanol.

DNA was extracted using the DNeasy® animal tissue QIAGEN kit (QIAGEN, Hilden, Germany) according to the manufacturer's instructions. Initially, universal primers were used to amplify a portion of the following genes: cytochrome *c* subunit I (LCO 1490 and HCO02198, *Folmer et al., 1994*) and 16S rRNA (Sar-L and Sbr-H, *Palumbi, 1996*). Primer-walking methodology involves the amplification of long overlapping PCR fragments of about 5–9 kb using specific outwardly facing primers. Those specific primers were designed over amplified fragments using the universal primers for 16S rRNA and COI described above. The design of the primers followed general rules: (1) melting temperatures of the primer pair between 55 and 65 °C and within 5 °C of each other, which is based on the GC content; (2) lengths from 18 to 30 nucleotides; (3) dinucleotide repeats (*e.g.*, ATATAT) were avoided, and (4) intra-primer homology to avert primer-dimer was also avoided. Each long PCR fragment was sequenced with Sanger technology, which produces sequences of about 700 bp. Over these sequences a new set of two inwardly facing primers is designed. The process of connecting fragments is repeated until reproducing the complete circular molecule (of about 16,000 bp for *Perna perna*). The list of the pairs of primers used in this study are in Supplemental information Table S1.

All PCR amplifications were conducted in 25-μl reactions containing 2.5 μl of 10X TaKaRa LA Taq™ Buffer II ($Mg^{2+}$ free), 2.5 mM $MgCl_2$, 2.5 mM of each dNTP, 0.5 μM of each primer, 1 μl template DNA (10 to 100 ng), and Taq DNA polymerase (5 units/μl, TaKaRa LA Taq). The PCR reactions had the following profile: denaturation period at 94 °C lasting for 30 s, followed by: 35 cycles at 98 °C for 10 s, annealing temperature specific to each primer combination for 30 s, extension at 68 °C of 60 s/1 kb; a final extension period at 72 °C for 10 min.

PCR products were cleaned using purified ethanol/sodium acetate precipitation and directly sequenced with the corresponding PCR primers. Sequencing was performed on an Applied Biosystems 3130xl Genetic Analyzer, using Sanger technology and the BigDye® Terminator v3.1 kit. Sequences of the complete mitochondrial genomes *P. perna* from this study were deposited in GenBank under the accession numbers OK576479 (Brazilian specimen B2); OK576480 (western South African specimen) and OK576481 (eastern South African specimen).

## Gene annotation, phylogenetic analyses and gene rearrangements

The approximate locations of the rRNA and protein coding genes were determined by aligning the obtained sequences against the mitochondrial genome of *Perna canaliculus* (NCBI GenBank accession number MK775558) using Geneious v. 7.1.4 (*Drummond et al., 2010*) with the option "highest sensitivity". Protein coding genes were confirmed by inferring open reading frames (ORFs), and by delimiting start and stop codons. "Cloverleaf" secondary structures of all tRNA genes were reconstructed manually upon localization in Geneious of the specific anticodons, and using the tRNAs of *P. canaliculus* as reference.

To reanalyze phylogenetic relationships within *Perna* and investigate the putative polyphyly of the genus as claimed in (*Uliano-Silva et al., 2016b*), we used a dataset (nine taxa, 14,251 bp) that included the newly sequenced genomes and the mitogenomes of the mussels *Perna viridis* (JQ970425), *P. canaliculus* (MG766134), *P. perna* (KM655841), *Brachidontes exustus* (KM233636) and *Musculista senhousia* (GU001954). Following the results of (*Uliano-Silva et al., 2016b*), we selected *Limnoperna fortunei* (KP756905) as the most appropriate outgroup. The deduced amino-acid sequences of each of the 13 protein-coding genes and the two ribosomal RNAs (12S rRNA and 16S rRNA) were aligned using Mafft v7.245 (Multiple alignment using Fast Fourier Transform) (*Katoh & Toh, 2010*) using the—auto option that automatically selects the appropriate strategy according to data size. Refinement of the alignment was performed "by eye" to maximize positional homology. The aligned sequences were concatenated into a single data set in interleaved Nexus format. The Akaike information criterion (*Akaike, 1973*) as implemented in jmodeltest-2.1.10 (*Darriba et al., 2012*; *Guindon & Gascuel, 2003*) selected the GTR+I+$\Gamma_4$ (general time-reversible substitution model with a proportion of invariant sites and among-site rate variation modeled by a discrete gamma-distribution with four categories) as the evolutionary model that best fitted the dataset.

The best partitioning schemes were identified by PartitionFinder2 v.2.1.1 (*Lanfear et al., 2017*) and used in maximum likelihood (ML) analysis. Our ML and BI analyses included *P. canaliculus*, *P. viridis* and four *P. perna* lineages (two specimens from Brazil and two from South Africa). *Musculista senhousia* and *Brachidontes exustus* were also included to compare with results from *Uliano-Silva et al. (2016b)*. We performed the ML analysis of the 13 protein-coding and the two ribosomal genes with RAxML v8.2.10 (*Stamatakis, 2014*) using the options – M that switches on estimation of individual per-partition branch lengths and – q which specifies the file name which contains the assignment of models to alignment partitions for multiple models of substitution under the GTRGAMMA (gamma model of rate heterogeneity). The best-scoring tree was determined from 100 randomized maximum-parsimony starting trees using the rapid hill-climbing algorithm. 500 bootstrap replicates were drawn on each best-scored ML tree using the exhaustive bootstrap algorithm.

Bayesian Inference was conducted with MrBayes v.3.2.7a (*Ronquist et al., 2012*) under the selected model (lset nst = 6 rates = invgamma;) using the same dataset as in ML

**Table 1 Comparison of the protein-coding and ribosomal gene sizes (in base pairs) between the three mitochondrial genomes newly sequenced.**

|  | *Perna perna* ESA | *Perna perna* WSA | *Perna perna* **Brazil (B2)** | *Perna perna* Brazil (B1) |
|---|---|---|---|---|
| Total size | 16,081 | 16,093 | 16,065 | 18,415 |
| l-rRNA | 1,114 | 1,118 | 1,118 | 1,185 |
| s-rRNA | 782 | 782 | 782 | 818 |
| atp6 | 714 | 714 | 714 | 509 |
| atp8 | 161 | 161 | 161 | absent |
| cytb | 1,151 | 1,151 | 1,151 | 1,157 |
| cox1 | 1,550 | 1,550 | 1,550 | 1,628 |
| cox2 | 692 | 692 | 692 | 704 |
| cox3 | 857 | 854 | 854 | 935 |
| ND1 | 935 | 935 | 935 | 905 |
| ND2 | 953 | 953 | 953 | 935 |
| ND3 | 353 | 353 | 353 | 350 |
| ND4 | 1,310 | 1,310 | 1,310 | 965 |
| ND4L | 266 | 266 | 266 | 260 |
| ND5 | 1,679 | 1,679 | 1,679 | 1,637 |
| ND6 | 476 | 476 | 476 | 536 |

Note:
Comparison of the protein-coding and ribosomal gene sizes (in base pairs) between the three mitochondrial genomes newly sequenced (highlighted in bold) and the *Perna perna* mitogenome (B1) published in (*Uliano-Silva et al., 2016a*). ESA, eastern South Africa; WSA, western South Africa.

analysis. The Metropolis coupled Markov chain Monte Carlo (MCMCMC) was sampled for $2 \times 10^7$ generations (two simultaneous MC chains) with a sampling frequency of 1,000.

We used CREx (*Bernt et al., 2007*) to determine the most parsimonious scenario for pairwise gene rearrangements, which included transpositions, reverse transpositions, reversals and tandem-duplication-random-loss (TDRL), given a phylogenetic hypothesis. The analysis was performed by applying the common intervals parameter for distance measurement. Based on the results from (*Lubośny et al., 2020*) and our ML analyses we consider the gene order of *Musculista senhousia* as the ancestral gene order for the genus *Perna*, and the gene order of *Limnoperna fortunei* as the ancestral gene order for *Brachidontes exustus* and the Brazilian specimen B1.

## RESULTS

The three *P. perna* mitogenomes sequenced in this study encode a total of 13 protein-coding genes, two rRNA (Table 1) and 23 tRNAs, one more (tRNA-Met, codon AUA) than usually found in metazoan mitochondrial DNA but already described for other bivalves (*Hoffmann, Boore & Brown, 1992*). The heavy strand encoded all genes. Overlapping of adjacent genes occurred between some of the tRNAs (Leu1–Asn; Ile–Gly) and between protein-coding genes (ND4–ATP8) in the three genomes. All the tRNAs of the three mitogenomes present a similar typical "cloverleaf" structure with four arms (acceptor, DHU, anti-codon and T Ψ C) except the two tRNA Ser that lack the DHU-arm, as

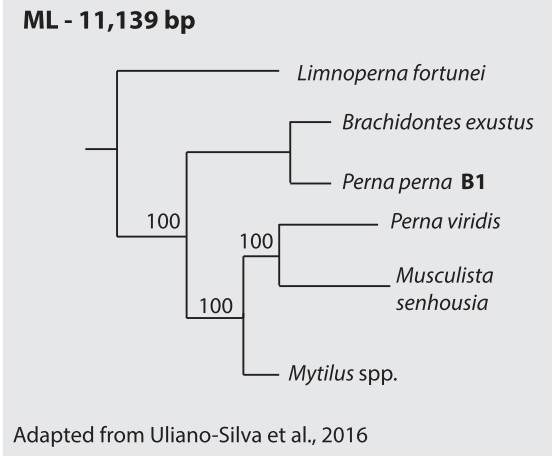

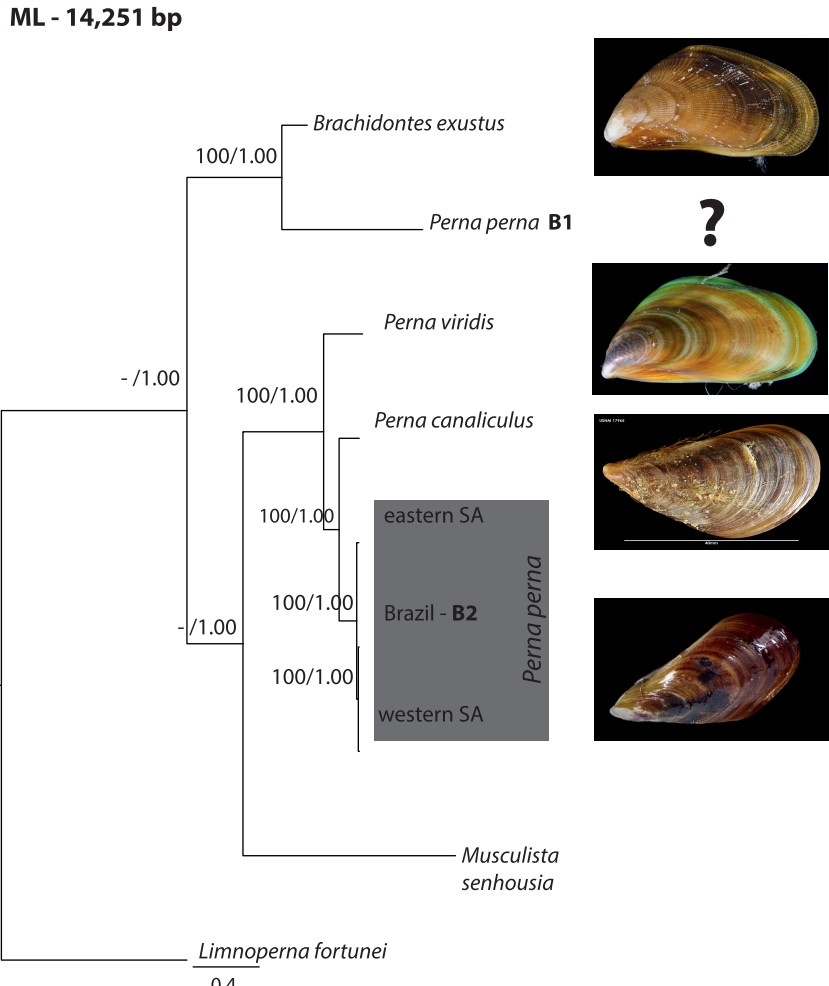

**Figure 1 Phylogenetic relationships of *Perna* and other Mytilidae based on complete mitochondrial genomes.** Phylogenetic relationships of *Perna* and other Mytilidae based on a maximum likelihood (ML) analysis of 12 mitochondrial protein-coding genes (ATP8 excluded) and the two ribosomal RNAs (12S and 16S) combined into an alignment of 14,251 base pairs. *Limnoperna fortunei* was selected as outgroup. *Perna perna* specimens sequenced in this study are highlighted in dark grey. Numbers at the nodes represent ML bootstrap proportions and BI posterior probabilities. Only values above 70% are depicted. B1 and B2: Brazilian *P. perna* samples sequenced in (*Uliano-Silva et al., 2016a*) and in this study, respectively. SA: South Africa. The inset highlighted in light grey shows a ML topology based on 11,139 bp corresponding to 12 mitochondrial protein-coding genes (ATP8 excluded) adapted from (*Uliano-Silva et al., 2016b*). Values at the nodes represent ML bootstrap proportions.

observed in nearly all metazoans (*Pons et al., 2019*). The sequences of 12 mitochondrial protein-coding genes (ATP8 was excluded because it is not present in *Musculista senhousia* and *P. perna* B1) and the two ribosomal RNAs from the analyzed Mytilidae were combined into an alignment of 14,251 nucleotide positions.

The ML (-ln L = 89440.34) tree shown in Fig. 1 exhibited an identical topology to the BI tree (-ln L = 90617.33). All *Perna* specimens clustered together except one of the Brazilian specimens of *P. perna* (B1) that grouped with *Brachidontes exustus* (Fig. 1). The western South African *P. perna* lineage grouped with our Brazilian *P. perna* (B2) and the eastern

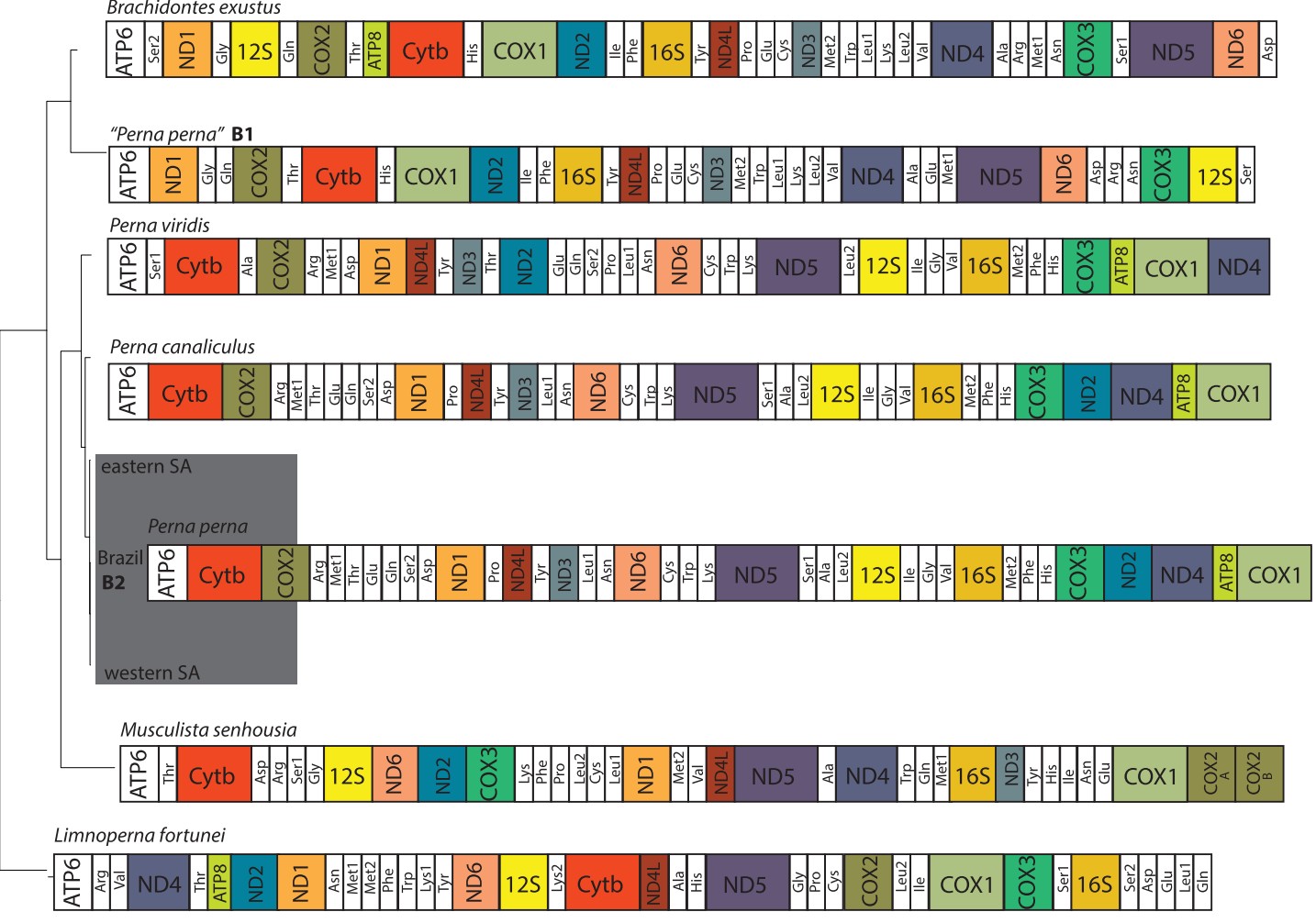

**Figure 2 Hypothesized mitochondrial gene order of the analyzed Mytilidae.** All genes are encoded by the heavy strand. *Perna perna* specimens sequenced in this study are highlighted in dark grey. B1 and B2: Brazilian *P. perna* samples sequenced in (*Uliano-Silva et al., 2016a*) and in this study, respectively. SA: South Africa. 

South African *P. perna* was retrieved as their sister lineage (Fig. 1). *Perna perna* (the three specimens sequenced in this study) and *P. canaliculus* were recovered as sister species.

Given previous results that suggested the non-monophyly of the genus *Perna* (*Uliano-Silva et al., 2016b*), we performed a nucleotide BLAST search (https://blast.ncbi.nlm.nih.gov/) using the cytochrome oxidase subunit I (COI) gene of the published Brazilian mitogenome B1 to find the closest matching sequences in NCBI GenBank database. COI is the most common gene for DNA barcoding and widely used for species identification. The BLAST search indicated that the COI from the Brazilian mitogenome of *P. perna* (B1) is 100% similar to *Mytilaster solisianus* (synonymized with *Brachidontes solisianus*) and there was no match with any other *Perna* species.

The two Brazilian *P. perna* mitochondrial genomes (B1 and B2) differed in size (Fig. 2). While our sample B2 (16,065 bp) is similar in length to the other two *Perna* species (*P. viridis*: 16,014 bp and *P. canaliculus*: 16,065 bp), B1 is 2,350 bp longer (18,415 bp).

The length of the mitogenomes of the eastern and western *P. perna* lineages were 16,081 and 16,093 bp, respectively. The gene ATP8 was only absent from the annotations of the Brazilian B1 *P. perna* and *M. senhousia* mitogenomes. We observed extensive differences in the gene order among the bivalve genera analyzed here (Fig. 2). Based on CREX analysis, the gene order of *P. viridis* evolved from its ancestral gene order (*M. senhousia*) as the result of a transposition of the block [ND3+Tyr] in which ND3 moved to the right and the tRNA Tyr to the left and three TDRLs (Fig. S1). The evolution of the gene order of *P. canaliculus/P. perna* eastern, western SA and Brazil B2 from its ancestral gene order (*P. viridis*) involved the following steps: (1) a transposition of the block [(ATP8+COX1) +ND4] in which (ATP8+COX1) moved to the right and the ND4 moved to the left; (2) the transposition of the ND2 that moved to the left; (3) two TDLRs (Fig. S2). According to our phylogenetic hypothesis, the gene order of the Brazilian specimen B1 evolved from the ancestral gene order of *L. fortunei* as the result of four TDRLs (Fig. S3). If we consider that the Brazilian specimen B1 evolved from the ancestral gene order of *M. senhousia* as the remaining *Perna*, we have to consider a transposition from the gene block (Ala+ND4) in which Ala moved to the right and ND4 to the left and four TDRLs (Fig. S4).

## DISCUSSION

Our phylogeny based on 12 mitochondrial genes (ATP8 excluded) and the two ribosomal RNAs showed that the western South African *P. perna* lineage groups with our Brazilian *P. perna* specimen B2. *Perna canaliculus* is retrieved as sister species of all *Perna* with the exception of the Brazilian specimen B1 (Fig. 1). A recent mitogenomic study analyzing mytilid mussels also found a sister relationship between *P. perna* and *P. canaliculus* (*Lubośny et al., 2020*). These results are in agreement with a previous study based on a fragment of the COI gene and the nuclear ITS1 (*Cunha et al., 2014*). In the phylogenetic analysis of *Uliano-Silva et al. (2016b)* also based on the same 12 mitochondrial protein-coding genes, the Brazilian *P. perna* B1 grouped with *Brachidontes exustus* and *P. viridis* with *Musculista senhousia* (Fig. 1-inset). Despite our larger taxon sampling (we included *P. canaliculus* and three more *P. perna* specimens in the analyses), the Brazilian *P. perna* sample B1 and *B. exustus* remained as sister species in all analyses (Fig. 1).

The well-established monophyly of the genus *Perna* (*Wood et al., 2007*) raises concerns regarding the taxonomic identification of the Brazilian *P. perna* specimen B1. The nucleotide BLAST search we performed indicated 100% similarity of the COI gene between the *P. perna* mitogenome B1 sequenced by *Uliano-Silva et al. (2016a)* and *Mytilaster* ("*Brachidontes*") *solisianus*. A misidentification of the specimen B1 could explain why it grouped with *B. exustus* instead of clustering with the remaining *P. perna*. We could not retrieve from the authors (*Uliano-Silva et al., 2016a*) any voucher picture of the specimen used to produce the mitogenome B1 that would enable us confirming its taxonomic classification. Further, a previous study using a chromosomal approach already pointed out that the COI from the Brazilian specimen was most likely incorrectly assigned to *P. perna* (*García-Souto et al., 2017*).

Gene order changes are usually associated with tRNAs due to their stem-and-loop structure (*Saccone et al., 1999*) and translocations involving protein-coding genes are

much rarer among metazoans (*Rawlings, Collins & Bieler, 2001*). Interestingly, the inferred rearrangements of the evolution of the gene order of *P. canaliculus* from its ancestral gene order (*P. viridis*) involved the transposition of the protein-coding genes ATP8, COX1, ND4 and ND2 (Supplemental Information S2). Given our phylogenetic hypothesis (Fig. 1), the inferred evolution of the gene order of *P. perna* B1 from its ancestral gene order (*L. fortunei*) involved the occurrence of four TDRLs (Supplemental Information S3). The alternative hypothesis of the evolution of the gene order of *P. perna* B1 from the ancestral gene order of the remaining *Perna* (*M. senhousia*) would imply four TDRLs and a transposition, which is less parsimonious and reinforces the idea of misidentification of B1 (Fig. S4).

The ATP8 is absent from the annotation of the Brazilian *P. perna* B1 mitogenome (Fig. 2). The absence of this gene may result from annotation errors due to its short and variable length (*Breton, Stewart & Hoeh, 2010*), which we believe is the case of *P. perna* B1, considering that ATP8 is present in its sister taxon *B. exustus*. However, we cannot fully discard its absence because there is no mitogenome available of *M. solisianus*, the species we consider correspond to the mitogenome *P. perna* B1. Regardless of the frequent gene rearrangements reported in bivalves, even at lower taxonomic levels (*Rawlings, Collins & Bieler, 2001*), and previous results suggesting independent origins for the eastern and western South African *P. perna* lineages (*Cunha et al., 2014*), we found no differences in the mitogenome gene order at the intraspecific level (Fig. 2).

The development of next-generation sequencing such as Illumina short-read sequencing, provides genome-wide data quickly and at relatively low cost. However, the existence of repetitive regions and missing data may complicate the assembly, even if a close reference is available (*Treangen & Salzberg, 2011*). The B1 mitogenome was produced with Illumina short-reads. We used the "primer-walking" methodology, which reduces the source of error due to the existence of larger overlapping fragments and the absence of repeats. The assembly of B1 was performed using as reference the *P. viridis* mitogenome (NC_018362), while we used *P. canaliculus*, which is the sister species of *P. perna*. We were not able to perform a new assembly of B1 using *P. canaliculus* as reference to analyze if gene order would change or to confirm that ATP8 was not present, as raw reads are not available in either public databases or from the authors.

## CONCLUSIONS

The previously published mitogenome of the brown mussel *Perna perna* from a Brazilian specimen (*Uliano-Silva et al., 2016a*) has a strikingly different gene order compared to its congeneric counterparts. Here, we used the primer-walking methodology to evaluate the unexpected gene order of this bivalve species. This time-consuming method produces very accurate results because of relying on sequentially-determined, overlapping fragments. We also inferred putative gene rearrangements at the intraspecific level using *P. canaliculus* as reference to facilitate annotation. The mitogenomes of the three *P. perna* specimens sequenced in this study (a second Brazilian specimen and the two South African lineages) showed the same gene order as the reference. Our results suggest a misidentification of the Brazilian specimen B1 used to produce the published mitogenome.

The non-monophyly of the genus *Perna* previously claimed by *Uliano-Silva et al. (2016b)* is not corroborated by our data.

## ACKNOWLEDGEMENTS

We are very grateful to Paulo A. Horta and Paulo R. Pagliosa for collecting the sample of *Perna perna* from Brazil and dealing with all the required permits. We also thank Luana Corona for helping in the laboratory. Phylogenetic analyses were performed on the R2C2 computational cluster facility provided by the IT Department of the University of Algarve and on the CCMAR computational cluster facility (CETA).

### Funding

Regina L. Cunha was funded by the transitional norm—DL 57/2016/CP1361/CT0013. This study received Portuguese national funds from FCT—Foundation for Science and Technology through project UIDB/04326/2020, and from the operational programmes CRESC Algarve 2020 and COMPETE 2020 through projects EMBRC.PT ALG-01-0145-FEDER-022121 and BIODATA.PT ALG-01-0145-FEDER-022231. This research was further supported by the Foundation for Science and Technology (FCT—MEC, Portugal (Grant Number: EXPL/BIA-BMA/0682/2021)), and the National Research Foundation of South Africa (Grant Number: 64801). The funders had no role in study design, data collection and analysis, decision to publish, or preparation of the manuscript.

### Grant Disclosures

The following grant information was disclosed by the authors:
DL 57/2016/CP1361/CT0013.
FCT—Foundation for Science and Technology Through Project: UIDB/04326/2020.
CRESC Algarve 2020 and COMPETE 2020 Through Projects: EMBRC.PT ALG-01-0145-FEDER-022121 and BIODATA.PT ALG-01-0145-FEDER-022231.
Foundation for Science and Technology (FCT—MEC, Portugal): EXPL/BIA-BMA/0682/2021.
National Research Foundation of South Africa: 64801.

### Competing Interests

Rita Castilho is an Academic Editor for PeerJ.

### Author Contributions

- Regina L Cunha conceived and designed the experiments, performed the experiments, analyzed the data, prepared figures and/or tables, authored or reviewed drafts of the article, and approved the final draft.
- Katy R Nicastro conceived and designed the experiments, authored or reviewed drafts of the article, and approved the final draft.
- Gerardo I Zardi conceived and designed the experiments, authored or reviewed drafts of the article, and approved the final draft.

- Celine Madeira performed the experiments, authored or reviewed drafts of the article, and approved the final draft.
- Christopher D McQuaid conceived and designed the experiments, authored or reviewed drafts of the article, and approved the final draft.
- Cymon J Cox analyzed the data, authored or reviewed drafts of the article, and approved the final draft.
- Rita Castilho analyzed the data, prepared figures and/or tables, authored or reviewed drafts of the article, and approved the final draft.

## Field Study Permissions

The following information was supplied relating to field study approvals (*i.e.*, approving body and any reference numbers):

The Federal University of Santa Catarina, Brazil approved sampling regarding the *Perna perna* specimen under the MTA-Material Transfer Agreement, SISGEN (registration number: RB435EFp).

The sampling of the two *Perna perna* specimens from South Africa was approved by Rhodes University under the project RES2019/30.

## DNA Deposition

The following information was supplied regarding the deposition of DNA sequences:

The sequences of the complete mitogenomes of *P. perna* produced in this study are available in NCBI GenBank: OK576479, OK576480 and OK576481 corresponding to the Brazilian, western and eastern South Africa samples, respectively.

## Data Availability

The complete mitochondrial genomes from three *Perna perna* specimens: western South Africa (W2); eastern South Africa (E2) and Brazil (B2) and the annotations from each of the three mitogenomes of Perna Perna are available in the Supplemental Files.

## Supplemental Information

Supplemental information for this article can be found online at http://dx.doi.org/10.7717/peerj.13953#supplemental-information.

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
