# Peer review of "Comparative mitogenomic analyses and gene rearrangements reject the alleged polyphyly of a bivalve genus"

_PeerJ, doi:10.7717/peerj.13953_

## Round 0.1 · original submission · Minor Revisions

In this manuscript the authors tackle the conundrum of an unusual gene order reported for a bivalve specimen from Brazil identified as Perna perna (B1). Molecular phylogenetic analysis of this specimen, along with others mt genomes, led the authors of the original study to conclude that the genus Perna was not monophyletic. The authors of this manuscript determined the gene order of three additional specimens of P. perna, one from Brazil, and two from South Africa, amplifying the mitochondrial genome in large chunks with long PCR, then primer walking through these fragments to determine the sequence, and finally inferring the gene order through analysis of the assembled contigs. I think the idea here is that the original B1 sequence was assembled from short Illumina reads, so that perhaps the apparent gene order differences were due to mis-assembly of short reads, or misled by duplicated genes, etc. If this were the case, the long fragments and primer walking strategy would be much less prone to these errors. In the final analysis, it looks like the main issue was that the B-1 specimen was misidentified, and is a specimen of a different genus, Brachidontes. Of course, this is not to say that the B-1 genome might not have had some assembly problems. There are some gene order differences between it and another published Brachidontes mt genome, so it remains to be seen whether these differences are real or assembly artifacts.

The two external reviewers were split on this manuscript, with one (reviewer 1) calling for minor revisions, and one (reviewer 2) for major revisions. There is an added complication in that reviewer 2 forthrightly and honestly revealed a potential conflict of interest. Given this, I support acceptance after revisions addressing reviewer’s comments, which should be addressed point-by-point in the letter accompanying the revised manuscript. I add a few comments below.

I agree with reviewer 1 that perhaps a little more effort should go into trying to determine what species B1 represents. The reviewer suggests Brachidontes solisianus.

I also agree with both reviewers that you need a better explication of your LA PCR and primer walking strategy. How about a figure with the amplified large fragments showing the size of the amplified products, their overlap, if any, the amplification primer locations, and walking primer locations. Finally, you suggest you use another species as a reference for assembly. Explain how this was done, given that you are primer walking large fragments.

You do not mention GenBank numbers for the new mt genomes in the manuscript. These need to be reported.

Reviewer 1 also notes that the permits are not listed for all samples used in the study.

Reviewer 2. notes that the annotated sequences would have been helpful for evaluating the manuscript. You did, however include the raw sequences in the supplementary files.

Reviewer 2 also notes the distinction between gene order differences which you observe, and gene order rearrangements, which you infer.

So, overall, a useful manuscript that addresses a confusing issue for those interested in these taxa.

Reviewer 1 ·

Basic reporting

There are various places where the language could be clarified or tightened up.

ABSTRACT
L35: “mollusks” for “molluscans”
L42: “... a phylogenetically more closely-related species.” This has no context in the abstract. It would be clearer to just name the species that you used — rather than providing the lengthy context.
L43: “because it relies”
L45: “lineages that do not share their most recent common ancestor” This is a point that is repeated throughout the paper. I find this phrasing awkward, and it isn’t particularly relevant to the objective of this study.
INTRODUCTION
L60: “among” for “within”
L64: Tempestini et al. 2020 — this is a paper about annelids, not mollusks
L80: fix parentheses with reference within parentheses
L87: same comment as with L45 above. This concept is actually described much more clearly in the title of Cunha et al., 2014
MATERIAL & METHODS
L114: “according to the manufacturer’s instructions”
L116-117: I don’t understand how the primers are presented in Table S1. They don’t seem to be paired. Even the primers for COI and 16S that are listed in the manuscript are not paired in Table S1. Only the forward primers seem to be listed. My apologies if I am missing something, but then I think that only makes the case that it can be clarified.
L126: “using purified ethanol/sodium”
L135: MK77555. This doesn’t refer to anything on Genbank. Isn’t it missing a digit?
L139: fix reference parentheses
RESULTS
L197: “indicated”
L198: those species names simply reflect a different combination, not synonyms. Also L220.
DISCUSION
L224: “We could not retrieve from the authors” State the reason.
L241: “The absence of this gene may result from annotation errors...” Is that the case here?
L250: “The B1 mitogenome was produced with Illumina short-reads.” That is clearly not the problem here. The issue described in this paper seems simply to be a misidentification.

Experimental design

no comment (but see validity of findings)

Validity of the findings

It is reported from the abstract and introduction that the impetus for this study was straightening out the confusion caused by a previous mitogenome for Perna perna known as B1. To solve this problem, the authors sequenced three additional Perna perna mitogenomes and performed phylogenetic analyses with various outgroup taxa to show that B1 is not part of the Perna perna species clade. Instead the protein-coding and ribosomal genes of B1 indicate a more recent common ancestor with a Branchidontes mitogenome. The B1 gene order is also similar to the available Branchidontes mitogenome. Also, BLASTing COI from B1 matches a known Branchidontes species (though one without a complete genome).

This seems to meet the objective set out in the introduction, although BLAST would have been a good place to start. Had the authors done so, it might have been more fruitful to spend time assembling a genome from a known specimen of Brachidontes solisianus.

Additional comments

I did not see any place in the manuscript where the authors intended to report the Genbank accession numbers for the genomes that they sequenced and annotated. The accession numbers provided by the authors with the review materials are not yet available on Genbank.

Permits are discussed L109-112. Only one of the collection agreements is provided in the supplementary materials.

Reviewer 2 ·

Basic reporting

Some references are missing.

Experimental design

Methods are not described with sufficient detail.

Validity of the findings

Some findings are not robust.

Additional comments

Dear Author,
Thank you for deciding to correct the error made by Uliano-Silva et al. six years ago. Unfortunately, you provided no data for evaluation and therefore I would normally not consider your paper for review. The reason for that is that the regular reader of the paper will have access to the data (sequences and annotations), and will be in much better position to then spot the potential problems which should have been caught by the reviewer. As a reviewer I therefore insist on having access to the same data which the reader of the published paper would have, including GenBank records containing the described mitogenomic data. However, since I am quite sure that your conclusion is largely correct I did review your work and the following are my comments, referenced by line numbers.

65: this is not entirely true. There is a case of apparent rearrangements within genus Mytilus which do qualify as such (doi:10.1007/s00239-010-9393-4)
66-70: This is a strange statement. "Primer walking" is not a method for directly accessing mitochondrial order of genes but simply a sequencing strategy. Once the sequence is correctly determined, annotation of genes follow and hence the resulting gene order is revealed. There is no direct connection between "primer walking" and "gene order". The fact that many mitogenomes were sequenced using this methodology (as per cited reference) is simply a consequence of the available sequencing methods (as this works best for Sanger sequencing of PCR or cloned DNA fragments).
71: why non-model organisms are brought in here? This techniques revolutionized mitogenomics of model organisms as well.
72: it is not clear why do you insist that these methods are "less expensive". They are in fact quite costly, it is impossible to produce any NGS result for a price of a single Sanger read. Perhaps you mean that the cost effectiveness of NGS based methods is better than that of the Sanger-based methods?
73-75: there is nothing intinsicly better in using Sanger reads instead of Illumina reads but their length. In fact, while the repeats are getting longer than the read length it is impossible to assembly the sequence. Please, do note that this is primarily an assembly problem, not directly related to the technology of obtaining the reads, as this seems to suggest.
78-83: Both assessments are based on a single mitogenomic announcement (properly cited), but the sentences suggest that these two aspects were studied separately. They were not. The second cited paper simply uses the data from the first one.
76-88: the paragraph misses the important paper on population genetics of Perna mussels, which shows different relationships between the species (doi:10.1016/j.ympev.2006.12.019) and is mainly based on a mitochondrial marker. It is trivial to obtain good barcoding type sequences from this paper and use it to assess the validity of the Uliano-Silva et al 2016a "Perna perna" mistake. In my opinion nothing else is needed.
89-92: this assessment is biased towards a very specific case. Unfortunately this is not true in a general case and is most likely irrelevant for the genus Perna. There are some methodological papers which must be considered in this context, if this is to be analyzed formally and not used as an anecdotal evidence. This always starts with a model, followed by some form of phylogeny reconstruction algorithm, with working implementation and followed by validation. The issue is far from solved, but apparently the short metazoan mitogenomes do not carry as much information as it seemed at the first glance, and does not allow meaningful phylogenetic conclusions, at least in general. A sequence of three tandem duplication random loss events may scramble the gene order beyond recognition. Some references to consider: doi:10.1007/978-3-540-87989-3_11, 10.1145/1109557.1109619, 10.1186/s12859-014-0354-6.
97: "SA" is not explained.
95-98: the sentence is unclear to me. What "extension" is mentioned here?
98-99: this sentence is not needed here, the exact span of the analysis is to be specified in Methods chapter.
109: since this is important to properly identify the original record, the most direct reference to the record deposited by Uliano-Silva in GenBank should be given here, not to the derived RefSeq accession number.
118: "manually" is a colloquial statement, essentially not needed at all, although some methodological details on primer design step would be nice.
120: don't use letter "X" to approximate "times" symbol. Why was the reaction buffer ten times more concentrated than recommended?
121: for long PCR projects which are then subject to sequencing it is mandatory to use a proofreading polymerase. Regular Taq is producing too many mistakes. The resulting reads may contain artifacts.
126: there is a grammar problem with this sentence, please check it out and correct.
132-133: I am not sure how the sequencing reads were eventually assembled, since there is no mention of it in the manuscript. If indeed these reads were just mapped onto the P. canaliculus mitogenome, then this is not acceptable as this would introduce unacceptable bias. I am almost sure the reads must have been somehow assembled first. Please provide the detailed account of your pipeline.
136-137: reconstructing "cloverleaf" secondary structures "by-hand" is meaningless. If these are to be mentioned, this must be done with proper modeling tools. There are plenty of such tools available (locarna, Vienna RNA package and many others).
139-143: the main conclusion of your manuscript is that Uliano-Silva made a mistake and published a different species' mtDNA as that of Perna perna. And yet, her manuscript is the only one you cite here, ignoring the papers you draw the data for your comparative analysis from. This is counterproductive and, at least for me, totally unacceptable. Moreover, the selection of just the sequences you list here is hardly justifiable as there are now many more Mytilid mitogenomes in GenBank, at least some of them with close relationship to the ones you did use.
147: refinement "by eye" is not only a colloquial phrase but also methodologically flawed approach. Please avoid doing so, because your results will not be reproducible otherwise.
156-159: this is seemingly duplication of the data given already at 140-143. However, it brings additional confusion. Each data set you analyze must be unequivocally defined in one place of the manuscript. This applies to all data sets analyzed, if in fact there was more than one (which is currently unclear).
170: please don't substitute letter "x" for "times" symbol. Also do not use mixed numbers, with the scientific notation the first number must not have trailing zeroes.
172-176: This paragraph is seemingly unrelated to the rest of this sub-chapter. Consider restructuring the methodology to find a better fit for this. However, this search is hardly something requiring a description, I am pretty positive this was done before by any reader of the Uliano-Silva unfortunate papers, with the obvious conclusion. Moreover, such conclusions were published several times already (for an example see doi:10.1016/j.jnc.2017.07.005). Unfortunately, the authors (of Uliano-Silva et al 2016b) did nothing about it. To be honest, I am not sure if this methodological approach needs an explanation in methodology at all.
180: second tRNA-Ser is a feature of all mitogenomes using translation table 5, I don't think there is anything unusual about it. Also, there is no need to cite the Hoffmann paper to justify its occurrence. However, to have 23 tRNAs there must be something more than that. In fact, maybe you are referring to the second rRNA-Met, the one with TAT anticodon? Well, its presence was indeed first postulated by Hoffmann and it was reported for other mytilids as well (it would be nice to have some more references here), so maybe this is the case? Please check this out.
181-182: the "additional" copies of tRNAs for Leu and Ser are a part of genetic code table 5 setup, the mitogenome can not possibly function without them, because Ser and Leu are encoded on two separate parts of the table which can not be served by a single anticodon. If any of them is missing, it would require explanation. Since they are not, there is no cause for highlighting this.
183-184: since the annotations were seemingly (according to the description given in Methods) copied from the closely related species, these are not your original findings. Or are they? This is unclear.
184-186, fig1: according to the description given in Methods, this picture was created manually, so presumably, these structures were imposed on the sequences at the drawing step. Therefore, this is a circular argument, unacceptable as such. Please provide the results of appropriate modelling, if you really feel the need to present these structures.
188: it is unclear which "specimens" lacked ATP8 gene.
189: why "eight"? Which ones?
191: do you mean identical topology? Or the branch lengths were also identical? I find it very unlikely. The cited figure contains just one tree - which one? The second one is pasted from Uliano-Silva et.al 2016b, which is neither needed nor entirely appropriate since you do not contest the phylogeny reconstruction, only the identity of the specimen. In the result section just your results should be presented.
197-199: yes, well, this should have been apparent at the times when this B1 was first described and published. This is not, strictly speaking your original finding.
200: fig 3 does not contain any information on the genome size, therefore this sentence is misleading.
202: mixing individual genomes with the notion that they represent "eastern and western lineages" is risky at best. Do you have any justification for inferring the existence of separate mitochondrial lineages in these phylogeographic context? Is this your assumption or result of some studies? Certainly you did not present them explicitly in the manuscript.
204-206: I do not agree with the assessment that we observe rearrangements. What we observe are differences in gene order. How did they came about is a matter of debate, but if there are to be any conclusions this must be set up in some coherent conceptual framework, not by arbitrary picking certain genes (and not the others). While the issue is far from clear, there are tools which actually help (CREx is one example).
206: the lack of ATP8 annotation is not the same as proof of its non-existence.
212: the repeated claim of "numerous rearrangements" calls for the proof: how many rearrangements were actually needed to account for the observed variability?
221: if that is so, then this paper should have been cited in the introduction. You did not have to sequence anything to arrive at this conclusion.
227-235: The reconstructions of phylogenetic trees within mytilids mitogenome data sets is not limited to the papers by Uliano-Silva. You should discuss your results also in the context of other published phylogenetic inferences. The few relevant examples include DOI: 10.1111/jzs.12354 (Fig 7), 10.1038/s41598-020-67976-6 (fig 3). Interestingly, this last one presents Perna perna in the same topological context as the one presented by you, but they apparently used some other ("unpublished" according to Table 2) Perna perna mitogenome, not the one from Uliano-Silva. This is worth investigating.
243: but Rawligs paper is about snalis?
250: citing NOVOPLASTY paper in this context is awkward since you did not use the software.
252: "absence of repeats"? The mitogenome of Perna perna does not have any significant repeats and does not pose serious problems for Illumina short reads assembly. Your implied critique is misdirected.
254-256: there is absolutely no reason to use reference-based assembly in any of these contexts.
259-260: six years ago is hardly "recently".

---

## Round 0.2 · Minor Revisions

In this revision, the authors have addressed the primary concerns of the reviewers, and the manuscript will be acceptable for publication pending the addition of a missing table. I was unable to find table S1, with the primer pairs used for sequencing. Please ensure it is included in the final version.

Reviewer 2 ·

Basic reporting

no comment

Experimental design

no comment

Validity of the findings

no comment

Additional comments

no comment

---

## Round 0.3 · accepted · Accept

The requested table has been added, so in my opinion, the manuscript is now acceptable for publication.